# Learning to Automatically Generate Accurate ECG Captions

**Mathieu G. G. Bartels**[1,2]                              MATHIEU.BARTELS@STUDENT.UVA.NL
**Ivona Najdenkoska** [2]                                        I.NAJDENKOSKA@UVA.NL
**Rutger R. van de Leur** [1]                              R.R.VANDELEUR@UMCUTRECHT.NL
**Arjan Sammani** [1]                                   A.ZABIHISAMMANI@UMCUTRECHT.NL
**Karim Taha** [1]                                            K.TAHA-2@UMCUTRECHT.NL
**David M. Knigge** [2]                                              D.M.KNIGGE@UVA.NL
**Pieter A. Doevendans** [1,3]                           P.DOEVENDANS@UMCUTRECHT.NL
**Marcel Worring** [2]                                                M.WORRING@UVA.NL
**René van Es** [1]                                          R.VANES-2@UMCUTRECHT.NL

[1] *Department of Cardiology, University Medical Center Utrecht, Utrecht, the Netherlands*

[2] *University of Amsterdam, Amsterdam, The Netherlands*

[3] *Netherlands Heart Institute, Utrecht, The Netherlands*

**Editors:** Under Review for MIDL 2022

## Abstract

The electrocardiogram (ECG) is an affordable, non-invasive and quick method to gain essential information about the electrical activity of the heart. Interpreting ECGs is a time-consuming process even for experienced cardiologists, which motivates the current usage of rule-based methods in clinical practice to automatically describe ECGs. However, in comparison to descriptions created by experts, ECG-descriptions generated by such rule-based methods show considerable limitations. Inspired by image captioning methods, we instead propose a data-driven approach for ECG description generation. We introduce a label-guided Transformer model, and show that it is possible to automatically generate relevant and readable ECG descriptions with a data-driven captioning model. We incorporate prior ECG labels into our model design, and show this improves the overall quality of generated descriptions. We find that training these models on free-text annotations of ECGs - instead of the clinically-used computer generated ECG descriptions - greatly improves performance. Moreover, we perform a human expert evaluation study of our best system, which shows that our data-driven approach improves upon existing rule-based methods.

**Keywords:** Transformer, Encoder-Decoder, ECG, Signal processing, ResNet, Captioning

## 1. Introduction

Computer vision methods have shown great improvements in image captioning, where a model is tasked to describe the content of an image using natural sentences (Vinyals et al., 2014). Such methods have also been successfully applied in the field of medical imaging, for example to obtain diagnoses from chest X-rays (Li et al., 2018; Chen et al., 2020; Najdenkoska et al., 2021; Hou et al., 2021) and histopathological scans (Ayesha et al., 2021). However, many other medical modalities, such as ECGs, are yet unexplored for captioning purposes.

Automated classification of ECG abnormalities using data-driven methods has recently reached expert levels (Ribeiro et al., 2019; Strodthoff et al., 2020). The next challenge in the automation of ECG diagnosis is the more challenging captioning task. A caption is directly interpretable, as it contains information that would aid experts in efficiently making a nuanced diagnosis, whereas a classification does not yield detailed insights for a certain diagnosis. Currently, descriptions generated by the rule-based Marquette system are used in clinical practise. This method requires improvements, as the generated descriptions are often incomplete (Schläpfer and Wellens, 2017). A good ECG description captures a variety of characteristics of the heart, like rhythm, axis and time intervals (De Jong et al., 2011). This implies that in contrast with free-text image captioning, where many different captions may be associated to any given natural image, an ECG caption is restricted in the range of plausible textual interpretations. Therefore, captioning of ECGs might be more feasible than for the other modalities. Recent approaches to medical image captioning have attempted to incorporate prior knowledge about the problem into the model design, aiding the model generalisation by explicitly including domain-specific human expert knowledge (Jing et al., 2018).

In this paper we aim to leverage the restricted domain of viable ECG captions, by constraining the captioning model through a prior based on human expert ECG descriptions. In doing so, we transfer medical image captioning methods to the yet unexplored domain of ECG captioning, by combining state-of-the-art (SOTA) image captioning architectures with SOTA architectures for classifying ECGs to generate cardiologist-level descriptions. To this end, we combine ECGnet (van de Leur et al., 2020) with a Transformer-based model (Vaswani et al., 2017), which takes into account ECG labels as prior knowledge.

This paper contains three contributions. First, our proposed data-driven approach is shown to significantly improve upon currently used rule-based methods. Second, we show that incorporating prior knowledge improves the quality of generated ECG captions. Third, we show that it is feasible to use the fully free-text annotations, instead of the corrected computer generated annotations, to train a captioning model.

## 1.1. Related Work

As we aim to generalize image captioning to ECGs we first review general image captioning methods, and from there medical report generation, followed by ECG classification and finally ECG captioning research.

**Image Captioning**  Image captioning is the task of describing visual contents with natural language. Successful image captioning models follow neural encoder-decoder architecture (Vinyals et al., 2014; Xu et al., 2015), where a convolutional neural network (CNN) is used as the encoder to learn meaningful image representations and a simple RNN as a decoder to generate a natural language description conditioned on the image. In other words, they are trained to maximise the likelihood p(S|I) where S is the sentence and I is an image.

To enhance this architecture, additional techniques are incorporated. For instance, Xu et al. (2015) extends the model by attending on the visual features at each timestep of the generation process. Anderson et al. (2017) propose to use region-based visual encoding of the images. More recent approaches use Transformers as a more powerful language

models for text generation (Cornia et al., 2020). In our ECG captioning approach, we follow neural encoder-decoder architecture by treating the ECG as a multi-channel signal.

**Medical report generation**     Medical report generation is regarded as translating a medical image into a textual report, fundamentally based on image captioning models. For instance, Jing et al. (2018) propose an approach for generating medical reports for chest X-rays. They propose a multi-task learning approach, to learn chest X-rays labels and generate a report with the same model. Their method allows to localise relevant image-regions and keep track of the overall topic of the report. For ECG captioning this approach may help confine the already limited generation space. Similarly to the improvement of image captioning models, medical report generation architectures are also enhanced with attention and Transformer-based models (Lovelace and Mortazavi, 2020; Chen et al., 2020; Najdenkoska et al., 2021; Hou et al., 2021; Wang et al., 2018). Existing work on medical report generation is focused on radiology reports, due to the availability of relevant datasets. Nevertheless, they represent a good example of text generation based on complex medical data with specific medial vocabulary, which is also applicable in ECG captioning.

**Classification of ECGs**   Current research combining ECGs and deep learning models, is mostly focused on classification of ECG findings. Recently, different signal encoders were tested on their ability to extract diagnostic information from an ECG (Strodthoff et al., 2020; Ribeiro et al., 2019; Hannun et al., 2019), also needed to interpret an ECG. This research suggests that the Resnet1d101 and Inception1d architectures scored best in classification tasks. The Resnet1d101 was used to classify a number of abnormalities, that compared to a cardiologist works at least as well. However, they do not have enough confidence that a deep neural net (DNN) outperforms humans. They state that DNNs cannot perform the same risk assessment as doctors do; are not good at handling rare diseases; and would not perform on unseen classes. These shortcomings also apply to ECG captioning. By generating ECG descriptions we try to improve the interpretabillity of automated ECG diagnoses, a focus in deep learning ECG methods (Vessies et al., 2021; Van De Leur et al., 2021).

**ECG captioning**     ECG captioning recently emerged as a promising direction in the quest of automating the report generation. However, no public dataset of sufficient quality ECG descriptions exist to benchmark. An existing work that approaches physician corrected descriptions using deep learning methods is Kashou et al. (2021), which uses a 1dResNet as an encoder and a Transformer as a decoder. Unfortunately, they do not mention a publicly available dataset for model comparison. Their research focuses on human evaluation of captions, originating from a rule-based, a data-driven method and a by physician-corrected. The authors validate their approach using expert evaluation. Since this metric is arguably most accurate, we employ expert evaluation to a limited extent as well. However, this is not a scalable solution to model evaluation; we would like to gain insight on model performance over a larger dataset. Therefore, we use the standard language and image captioning metrics like BLEU, METEOR, ROUGE and CIDEr (Papineni et al., 2002), (Denkowski and Lavie, 2014), (Lin, 2004), (Vedantam et al., 2014). Their approach refined 66 possible tokens that describe an ECG, and see a description as a series of those tokens. This contrasts with our approach which uses free text with a much larger vocabulary size.

## 2. Methods

### 2.1. Problem formulation

Our approach is inspired by neural encoder-decoder architectures for image captioning. We choose to employ this framework as we hypothesize it generalises well to the current problem setting; our input data bears great similarity to image data in that we can treat the ECG as a multi-channel bounded signal with spatial locality containing regions of interest.

Our model takes as input a raw ECG signal and encodes it into feature representation of fixed dimensions. Then, the decoder uses this representation to generate the ECG description in an auto-regressive manner. Thus, the objective function that maximizes the probability of generating a certain caption given an ECG is formulated as follows:

$$\theta^* = \arg\max_{\theta} \sum_{E,S} log(p(S|E,\theta)), \tag{1}$$

where $\theta$ are the parameters of the model, E is the ECG, and S is the correct ECG caption with a predefined maximum length of 50 tokens. The probability of the caption S is factorised using the chain-rule, as the product of the probability of the individual words $S_1, \ldots S_n$ where $n$ is the length of the current caption, shown in Equation 2:

$$log(p(S|E,\Theta)) = \sum_{n} \log p(s_n|E,\theta,s_1,s_2,\ldots,s_{n-1}) \tag{2}$$

To solve this formulation, we use appropriate neural networks to define the encoder and decoder in the next section.

### 2.2. Models definition

Although our problem bears similarity to image captioning, we argue there is a distinction which we aim to leverage to improve viability of the generated ECG captions. Whereas in image captioning the goal is to generate free textual descriptions for images in a wide variety of settings, our work only aims to generate clinically accurate textual descriptions of ECGs. As described in section 1, an accurate ECG description contains information on a number of characteristics of the input signal. Intuitively, this greatly restricts the range of plausible realistic textual descriptions that may be associated to any single ECG. We assume that embedding such information specific to this problem into our model may greatly improve the generated ECG descriptions.

**Prior-Knowledge-ECGnet-LSTM** To this end, we introduce our baseline model, denoted as Prior-Knowledge-ECGnet-LSTM (PKEL), illustrated in Figure 1(a). This model contains the ECGnet (van de Leur et al., 2020), a 37-layer 1D Residual CNN as an encoder and a top-down attention LSTM (Anderson et al., 2017) as a decoder. This model has an intermediate classification step, by using a simple classification layer on the encoded ECG representation. This classification step yields the labels, which are encoded with a learnable embedding layer. The addition of the classification step adds an extra term to the objective function shown in Equation 3. The $\lambda$ terms are weighting hyperparameters that sum to one. In our experiments, we set these hyperparameters to $\lambda_1 = 0.7, \lambda_2 = 0.3$ where

$\lambda_1$ is set to be larger than $\lambda_2$.

$$L(E,S) = -\lambda_1 \sum_{t=1}^{N} log(p_t(s_t)) + \lambda_2 l_{tag} \tag{3}$$

Inspired by Jing et al. (2018) we apply co-attention layer to combine the ECG representation and the learned semantic labels. The output of the co-attention is denoted as a context vector and is used to initialise the decoder part. The decoder consists of two LSTMs and a so-called signal attention module which learns to focus on a relevant part of the ECG representation when generating the t-th word.

**Prior-Knowledge-Transformer** We improve the baseline by relying on Transformers (Vaswani et al., 2017) as a more powerful model to capture long-term dependencies in sequences. Firstly, we add a Transformer encoder to learn a revised representation of the ECG, similarly as in Najdenkoska et al. (2021). The classification module to obtain the labels stays the same as in the PKEL model. Secondly, the context vector produced by the co-attention module, is used together with the encoded ECG representation to initialise the Transformer decoder, which generates the ECG caption word-by-word. The resulting model is named Prior-Knowledge-Transformer (PKTransformer), illustrated in Figure 1(b).

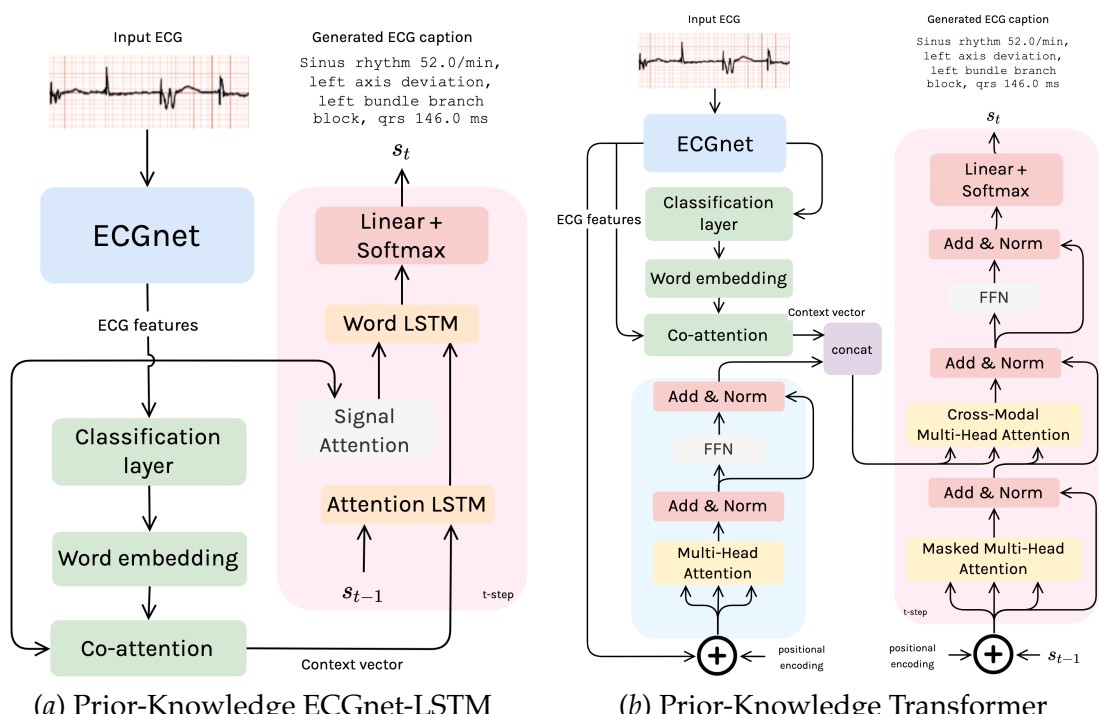

(a) Prior-Knowledge ECGnet-LSTM      (b) Prior-Knowledge Transformer

Figure 1: Architecture design using the prior-knowledge modules

## 3. Experiments and Results

### 3.1. Data description

The datasets used for this research are two private datasets obtained by the University Medical Center Utrecht (UMCU). The study was conducted under a protocol approved by the UMCU Institutional Review Board using a waiver of written informed consent. The first, the physician-corrected dataset, contains 368.430 ECGs with Dutch descriptions as they are used in the clinic, first generated by the Marquette system and then corrected by a physician. The second dataset, the physician-annotated dataset, includes 68.208 ECGs with Dutch descriptions fully made by physicians as part of their clinical routine, not being biased by seeing rule-based descriptions first, making them the golden standard. The two datasets are split into training, validation and test set in a 70/20/10 split, if a patient had multiple ECGs they only appear in one split. For the pre-processing of the physician-corrected dataset, textual references to previous ECGs are removed, although the references have diagnostic value, our approach only aims to describe a single ECG. The physician-annotated is more heterogeneous as it is free-text created by humans. Moreover, it contains an extensive range of numbers that individually sparsely occur. When tokenized, the numbers lose their ordinal properties, complicating the learning of good representations. There are sophisticated methods to deal with numbers in text generation. However, the pragmatic approach used for this paper was to tokenize the numbers and use a rule-based system to replace these tokens afterwards. The tokens are identified using regex on the words surrounding the numbers. Five tokens are introduced, 1) the ventricular rate, 2) the heart axis in degrees, 3) the PQ-interval, 4) the QRS-interval, and 5) the QTc-time. This approach reduced the vocabulary size while maintaining meaningful generations.

The different topics and gathering of the labels are described in the paper by van de Leur et al. (2020). They tokenized the ECG descriptions and unified the tokens with the same meaning. The topic are extracted from the physician-corrected labels and are not available for the annotated dataset.

### 3.2. Quantitative Evaluation

The two architectures described in Section 2 are evaluated with commonly used evaluation metrics for natural language generation (NLG); BLEU, METEOR, ROUGE and CIDEr. During the decoding process, the best word sampling method was empirically found to be greedy sampling and is used for all the experiments. The performance is shown in Table 1 together with the evaluation scores of the rule-based method. For the physician-annotated dataset, the encoder was pre-trained on the physician-corrected dataset and frozen for the first five epochs.

We find three major results in this experiment shown in Table 1. Firstly, using the prior knowledge in the PKEL and PKTransformer models increases performance on the metrics. Secondly, enriching the representational capacities of the encoder using either the prior labels or a Transformer encoder, improves performance in the current ECG captioning task. Thirdly, the rule-based system outperforms our models on the physician-corrected dataset, but is not close to our models on the physician-annotated dataset.

Table 1: Quantitative evaluation of the different architectures on the physician-corrected and physician-annotated dataset. The Prior knowledge column indicates whether the Prior knowledge is module is used or ablated.

| Dataset | Source | Prior labels | BLEU1 | BLEU2 | BLEU3 | BLEU4 | METEOR | ROUGE_L | CIDEr |
|---|---|---|---|---|---|---|---|---|---|
| physician-corrected | PKEL | ✓ | **0.458** | **0.391** | **0.343** | **0.304** | 0.264 | **0.639** | 2.535 |
| | | ✗ | 0.436 | 0.360 | 0.308 | 0.266 | 0.247 | 0.615 | 2.275 |
| | PKTransformer | ✓ | 0.454 | 0.388 | 0.341 | **0.304** | **0.265** | **0.639** | **2.546** |
| | | ✗ | 0.418 | 0.353 | 0.308 | 0.271 | 0.244 | 0.624 | 2.418 |
| | Rule-Based | N/A | **0.670** | **0.620** | **0.585** | **0.555** | **0.402** | **0.717** | **3.825** |
| physician-annotated | PKEL | ✓ | 0.375 | 0.284 | 0.225 | 0.174 | 0.213 | 0.373 | 0.618 |
| | | ✗ | 0.290 | 0.214 | 0.167 | 0.125 | 0.182 | 0.328 | 0.324 |
| | PKTransformer | ✓ | **0.382** | **0.291** | **0.231** | **0.179** | **0.220** | **0.373** | **0.640** |
| | | ✗ | 0.373 | 0.287 | 0.228 | 0.178 | 0.216 | 0.372 | 0.581 |
| | Rule-Based | N/A | 0.022 | 0.007 | 0.002 | 0.001 | 0.019 | 0.050 | 0.045 |

The first finding confirms our hypothesis that adding prior knowledge helps generations. The second finding shows the power of the Transformer architecture. We conjecture possible reasons for this finding are: 1) The usage of multiple attention heads allows to encode multiple relationships and nuances for each word that is generated; 2) The Transformer model is used in many multi-modal tasks to achieve SOTA results (Lu et al., 2019; Tsai et al., 2019), thus it is expected that the Transformer model achieves good results on this task as well. The third finding, can be explained when looking at the origin of the dataset, the ground-truth is the caption used in the clinic, which is made by a doctor who first reads the generation made by the rule-based system and adjusts it to an improved ground-truth caption, heavily correlating the two descriptions. For the physician-annotated dataset, the rule-based method performed poorly, since the expert descriptions are more eleborate than the rule-based method.

### 3.3. Qualitative Evaluation

In the qualitative experiment, two cardiologists, working in the cardiology outpatient clinic, were asked to evaluate ECG descriptions. They evaluated 30 ECGs randomly selected from the test set. Each of these ECGs were interpreted by a expert, the PKEL model trained on the physician-annotated dataset, the PKEL model trained on the physician-corrected dataset, and the rule-based system. The ECG-description pairs were shown in a random order to prevent bias. Experts rated the description with the following labels: 1) good, 2) small adjustments needed, 3) large adjustments needed, and 4) unusable. Results, visualised in figure Figure 2, show that the PKEL model outperforms the rule-based approach in the qualitative evaluation. The cardiologist interpretations are superior to our model. The cardiologist interpretations are good in only 61% of the cases, showing a large inter observer variance. We additionally show a number of generated descriptions in Table 2. In conclusion, these results show that our data-driven approach generally outperforms rule-based systems in generating expert-level ECG descriptions.

### 4. Conclusion

This paper showed that it is possible to automatically generate accurate ECG descriptions with a data-driven model, allowing for better interpretability. Our design choices, such

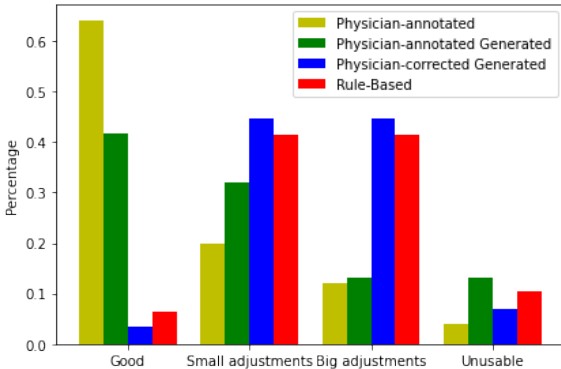

Figure 2: Expert evaluation of the ECG descriptions. The height of the bar represents what percentage of labels was assigned to the class displayed on the x-axis. From left to right the descriptions originate from: cardiologist description, PKEL model trained on the physician-annotated dataset, PKEL model trained on the physician-corrected dataset and the rule-based system

Table 2: ECG descriptions by an physician, our model and the rule-based system. The columns contain: the source of the generation and the translated generations, the ECGs and original Dutch generations are found in Figure 5, 6, 7 and Table 3

| ECG | Source | English translation |
|---|---|---|
| Figure 5 | physician-annotated | sinus rhythm 52 /min, left axis deviation, left bundle branch block |
| | PKEL generated | sinus rhythm 52.0 /min left axis deviation left bundle branch block qrs 146.0 ms |
| | Rule-Based | sinus bradycardia, left axis deviation, left bundle branch block |
| Figure 6 | physician-annotated | sinus rhythm 75 /min normal QRS axis normal conduction normal R wave progression normal repolarisation |
| | PKEL generated | sinus rhythm 72.0 /min normal QRS axis and normal conduction no pathological Q waves no ST deviation, normal repolarisation |
| | Rule-Based | sinus rhythm |
| Figure 7 | physician-annotated | sinus bradycardia 54 /min, normal QRS axis PQ 0.23 sec QRS 0.10 sec QS complexes V1-V3 persisting ST elevation V1-V4 normal T-waves QTc 0.40 sec |
| | PKEL generated | sinus rhythm 54.0 /min, normal QRS axis, PQ 0.230 sec, QRS 94 ms, QS complexes V1-V3, QTc 0.396 sec |
| | Rule-Based | sinus bradycardia with first-degree AV block anterior wall infarction of which the age is unknown |

as incorporating labels as prior knowledge and enriching the encoder, show to improve the overall quality of generated descriptions. Our findings show that training on free-text annotations of ECG descriptions improves model performance, compared to training on a set of clinically-used ECG descriptions. Moreover, we perform a human evaluation study on one of the systems by consulting cardiologists, indicating that it is possible to use the ECG descriptions generated by data-driven models over the rule-based ones. Lastly, these results may prove relevant to the field of image captioning; our findings show that we can greatly improve the quality of generated captions by incorporating prior knowledge on the semantic content of natural texts relevant to the problem setting.

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

**Appendix A.**

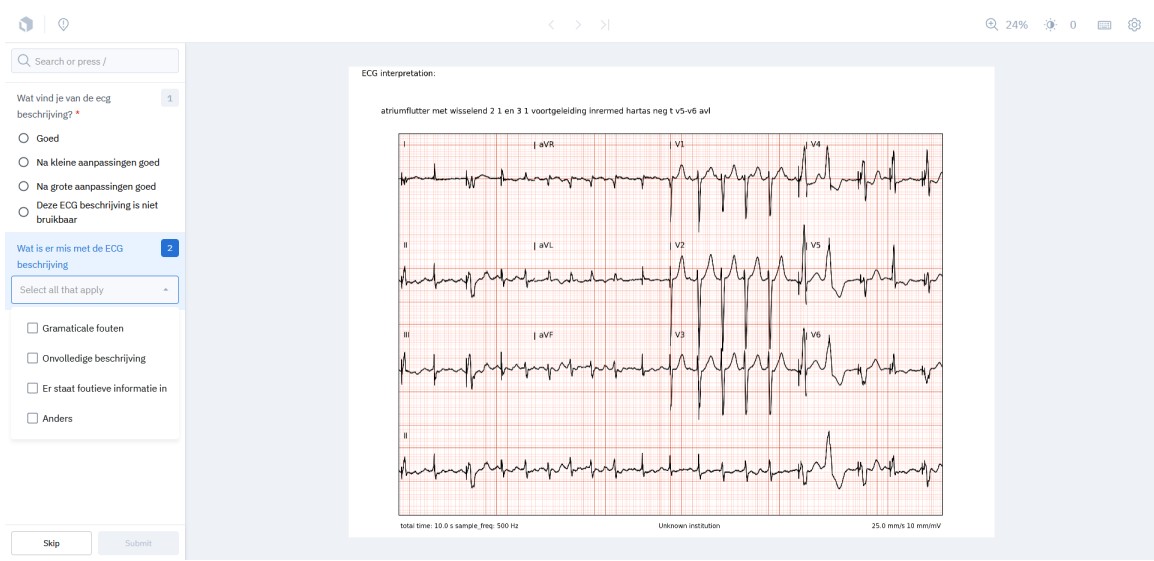

Figure 3: Layout of the qualitative experiment. On the left side the participant can select what they think of the ECG description that is displayed above the ECG. Any imperfection in the description can be selected on the bottom-left.

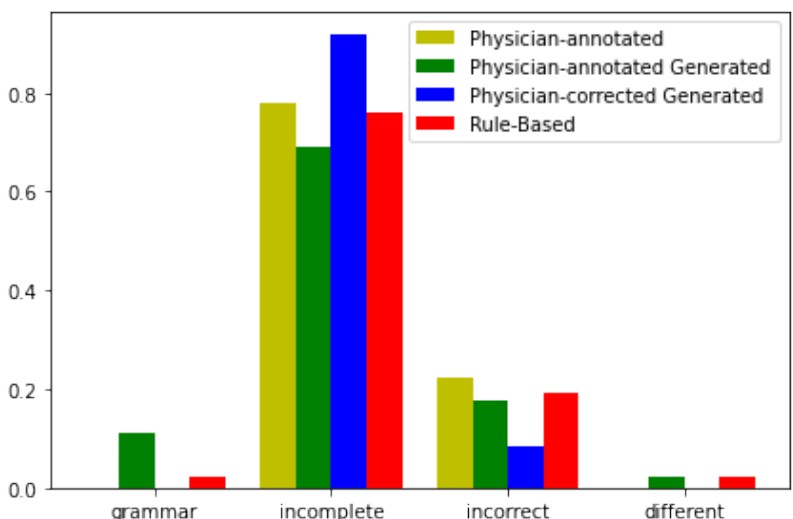

Figure 4: In this figure the two experts express what is wrong with the ECG description when it doesn't receive a good score

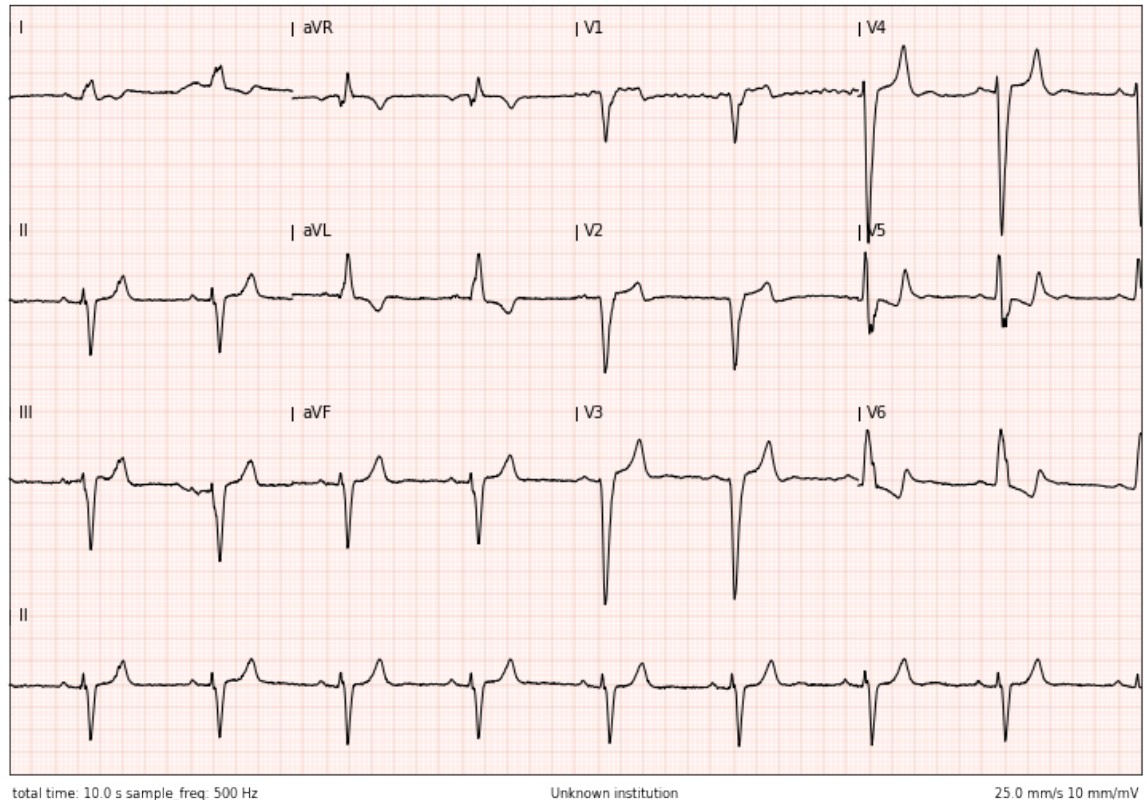

Figure 5: ECG images that accompanies Table 2, row 1-3. Sinus rhythm 52.0/min, left axis deviation, left bundle branch block, qrs 146.0 ms

Table 3: ECG descriptions by an physician, our model and the rule-based system. The columns contain: the source of the generation and the original Dutch generations, the corresponding ECGs are found in Figure 5, 6 7, This table complements Table 2

| Source | Dutch |
|---|---|
| Expert | sr 52 /min linkeras lbtb |
| PKEL generated | sr 52.0 /min linker as lbtb qrs 146.0 ms |
| Rule-Based | sinusbradycardie linker asdeviatie linker bundeltakblock |
| Expert | sr 75 /min intermediaire hartas normale \\geleiding normale r progressie normale repolarisatie |
| PKEL generated | sr 72.0 /min normale as en geleiding geen pathologische q s geen st deviatie normale repolarisatie |
| Rule-Based | sinusritme |
| Expert | sb 54 /min int as pq 0.23 sec qrs 0.10 sec qs v1-v3 persisterend st elevatie v1-v4 normale t-toppen qtc 0.40 sec |
| PKEL generated | sr 54.0 /min int as pq 0.230 sec qrs 94 ms qs v1-v3 qtc 0.396 sec |
| Rule-Based | sinusbradycardie met 1e graads av-block voorwandinfarct onbepaalde leeftijd |

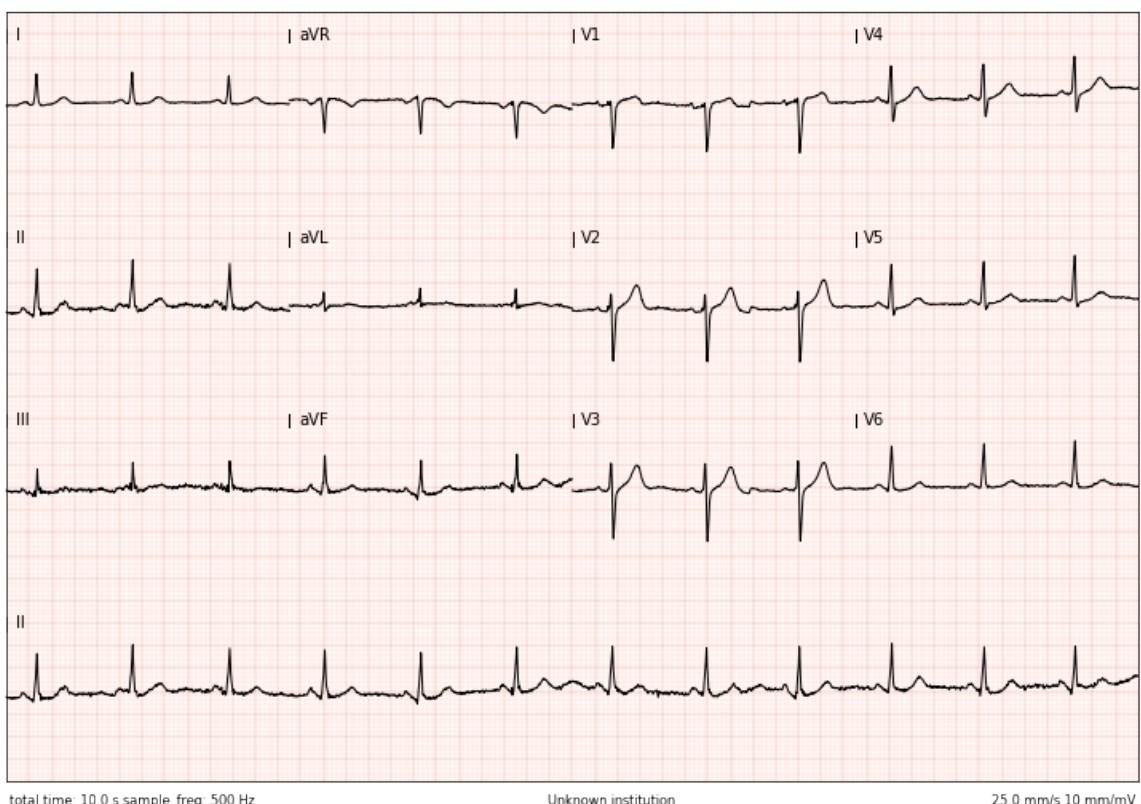

Figure 6: ECG images that accompanies Table 2, row 3-6. Sinus rhythm 72.0/min, normal QRS axis and normal conduction, no pathological Q waves, no ST deviation, normal repolarisation

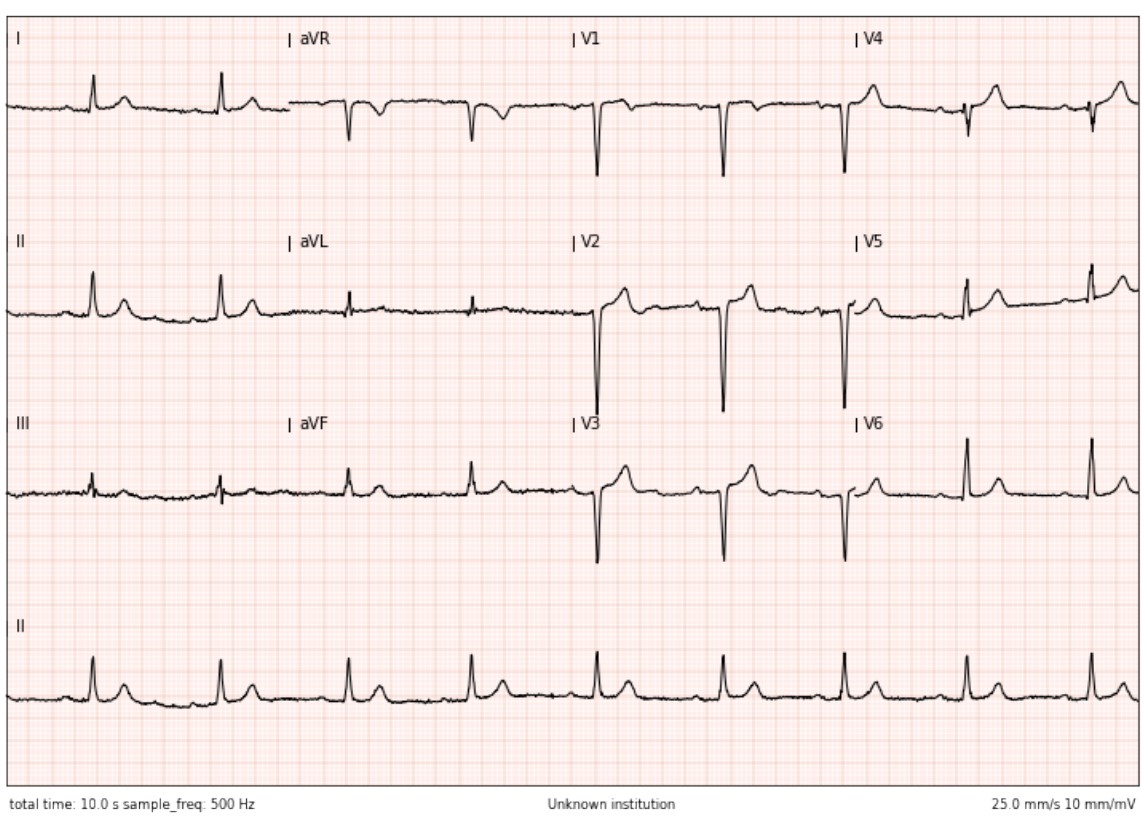

Figure 7: ECG images that accompanies Table 2, row 6-9. sinus rhythm 54.0/min, normal QRS axis, PQ 0.230 sec, QRS 94 ms, QS complexes V1-V3, QTc 0.396 sec

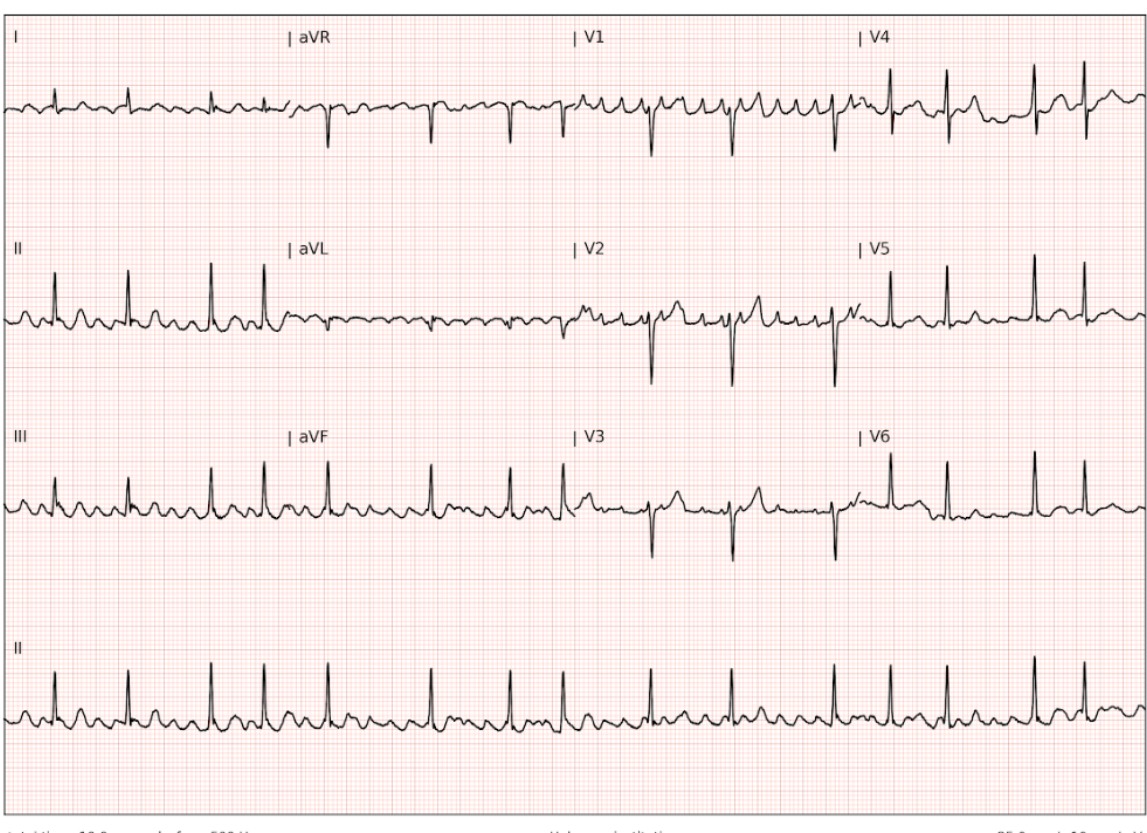

Figure 8: Example ECG with description: atriumflutter met wisselende volgfrequentie intermediaire hartas qrs 0.07 sec qtc niet verlengd geen pathologische q s geen st-segment afwijkingen normale t-toppen

Table 4: This table shows the inter-rater variability between the two experts for each of the generation sources. The higher the Pearson R, Cohen's kappa and Krippendorff's alpha values, the more agreement the two experts had on the quality of the description in Figure 2.

| metric / generation source | Pearson R | Pearson R p-value | Cohen's kappa | Krippendorff's alpha |
|---|---|---|---|---|
| physician-annotated | 0.563 | 0.003 | 0.315 | 0.419 |
| physician-annotated generated | 0.436 | 0.01 | 0.37 | 0.41 |
| physician-corrected generated | 0.323 | 0.054 | 0.045 | 0.304 |
| Rule-Based | 0.225 | 0.25 | 0.118 | 0.27 |

