# OpenReview forum: " Learning to Automatically Generate Accurate ECG Captions"
_MIDL.io/2022/Conference — MIDL 2022_

### Official Review · Reviewer_HC86 · 2022-01-21

**Confidence:** 5
**Preliminary Rating:** 3
**Recommendation:** Poster

**Summary:**

The paper proposed to apply image captioning methods for ECG description generation. The method has been evaluated quantitatively using BLEU scores and qualitatively by two cardiologists. A combination of Transformer architectures is used to incorporate prior knowledge about the expected annotations.
It is questionably if image analysis is the right approach for this type of data.

**Strengths:**

- the amount of data seems to be sufficient for such a task
- the paper is reasonably well written
- evaluation seems to be sound and results seem good
- the method has been studied on several levels

**Weaknesses:**

- clinical usability unclear
- using raw ECG data would be preferable
- limited methodological insights, limited discussion about the stretched term 'prior knowledge' in the specific context of this paper
- some missing background work

**Deanonymize Review:**

no

**Detailed Comments:**

It is hard to tell clinical usability given that only two cardiologists evaluated 30 generated reports qualitatively. Figure 3 probably would benefit from a slightly different presentation to make it easier to read.

- This is clearly interesting research but I guess the conference is called medical imaging with deep learning and not medical data with deep learning. Thus, I am unsure how ECG data evaluation fits MIDL. Along this line I would also assume that captioning the image is not quite the right approach here. Wouldn't it be more effective to caption the raw ECG data here?

- methodological insights beyond image captioning are limited. I don't see what special changes the authors had to made to the proposed transformer models.

- clinical report generation has also been explored in the context of chest X-Ray reporting, e.g.
Wang X, Peng Y, Lu L, Lu Z, Summers RM. Tienet: Text-image embedding network for common thorax disease classification and reporting in chest x-rays. InProceedings of the IEEE conference on computer vision and pattern recognition 2018 (pp. 9049-9058).
Hou B, Kaissis G, Summers RM, Kainz B. RATCHET: Medical Transformer for Chest X-ray Diagnosis and Reporting. In International Conference on Medical Image Computing and Computer-Assisted Intervention 2021 Sep 27 (pp. 293-303). Springer, Cham.

**Final Rating After The Rebuttal:**

4: Weak Accept

**Justification Of The Final Rating:**

Thank you for addressing the points raised above and adding clarifications to the presented work. ECGs from prints are still a questionable application. This multi-1D signal is not really comparable to other modalities since there is a linear relationship between signal and 'image'. Other modalities exhibit more complex signal to image mappings and thus require the image as representation. ECG does not. it can be interpreted as is. Anyway, since the other reviewers and the authors have good arguments it could maybe be accepted.

**Paper Type:**

methodological development

**Questions To Address In The Rebuttal:**


Why wouldn't one want to use ECG raw data instead of images. The images are clearly a side-product of the current clinical workflow and not an accurate representation of the underlying data.

Please comment on your key methodological contributions? What makes your approach work in comparison to just applying a random BERT model apart from the 'prior knowledge' language bias?

**Special Issue:**

no

---

### Official Review · Reviewer_gmtd · 2022-01-22

**Confidence:** 4
**Preliminary Rating:** 5
**Recommendation:** Oral

**Summary:**

This work presents a framework to generate descriptions from ECGs. The current SOTA approach to this is the use of rule-based systems. The work takes inspiration from encode-decoder architectures for image captioning and it uses a transformer model that encodes prior ECG labels knowledge.The latter is used to restraint the type of text/descriptions that are generated as, differently from image captioning, the text to be generated is not "free text".

The work is evaluated in a set of private datasets showing good results both quantitative and qualitative.

**Strengths:**

- This is an interesting and well grounded paper that addresses a novel application for the MIDL community. I consider it is worth of presenting at the conference as it brings new and fresh ideas that are worth discussion.
- The proposed method reports results which improve SOTA rule-based approaches
- Sound formulation
- Good experiments and analysis

**Weaknesses:**

- one could argue that ECGs are not really images. However, ECGs have been historically considered as part of the medical imaging communities
- The datasets are private
- The paper is good. The weaknesses are minor

**Deanonymize Review:**

no

**Detailed Comments:**

- The classification module to obatain -> obtain
- The Prior knoledge  -> Knowledge
- Please revise the paper carefully to make sure no other typos are there
- Are you using raw signals or images? This is not fully clear from the paper

**Final Rating After The Rebuttal:**

5: Strong Accept

**Justification Of The Final Rating:**

My  opinion about this paper remains the same. I consider it presents novel ideas that are definiively worth of discussion at MIDL. For this reason, I recommend acceptance and an oral presentation at the conference.

**Paper Type:**

methodological development

**Questions To Address In The Rebuttal:**

The paper states that "we show that it is feasible to use the fully free-text annotations". However, this stage requires the intervention of a rule-based system (Sec 4.5). Could you comment on this?  Perhaps this claim should be softened?

**Special Issue:**

yes

---

### Official Review · Reviewer_Mwgo · 2022-01-25

**Confidence:** 4
**Preliminary Rating:** 5
**Recommendation:** Best Paper Award, Oral

**Summary:**

The paper presents a captioning model for ECG data. The data collection and labelling done for the work is substantial and rich. The models themselves are simple but effective. The introduction of the number tokenisation strategy is really interesting and appears to work very well, and the results show a significant improvement over prior work. Overall, this is an excellent paper.


**Strengths:**

. Strong data collection and labelling process, including automated+corrected and human-only labeled data.
. Simple but effective methods for ECG captioning
. Strong validation of the output using both quantitative and human-led qualitative approaches


**Weaknesses:**

. It uses a proprietary dataset, meaning that the results cannot be reproduced or improved upon by other teams
. A stronger ablation study and hyper-parameter optimisation could have significantly improved overall performance
. The paper could have benefited from some degree of inter-rater variability study as it seems that clinicians disagree with each other’s reports.
. A more statistical analysis of the quantitative scores could have helped.

**Deanonymize Review:**

no

**Detailed Comments:**

. Readers not versed on ECG annotations will not understand the differences between the physician-corrected and physician-annotated system and why one approach is better than another.

. “references to previous ECGs are removed” what kind of references? If these are contextual (e.g. “the  pattern is the same as the previous ECG”) they cannot be temoved as one would also be removing information. The text would have to be edited to include this contextual information without using a reference to a previous ECG.

. The tokenisation of the numbers is a smart approach, but how were these tokens identified in the first place? Was it done by a concept identification module? Or was it done manually? Given the numbers of datasets, doing it manually seems quite time consuming.

. In section 4.3, can you confirm the same 30 ECGs were assessed for all methods? If so, was the order of the methods randomised, ie did the clinicians always saw the output of one model first for all ECGs? This can introduce biases, as the clinicians can change they way they read a piece of text depending on what they have read before.

. Would be good to see the results of fig 3 for the PKT model.


**Final Rating After The Rebuttal:**

5: Strong Accept

**Justification Of The Final Rating:**

The authors have addressed most of my points in full, and I continue to believe this is a great paper and will be of benefit to the MIDL community. I would only add the suggesting to replace the numerical value tokenisation from using regular expressions to using concept identification models, as I suspect that such tokenisation approach might be non optimal, primarily if trained on broader data from more sites.

**Paper Type:**

both

**Questions To Address In The Rebuttal:**

I personally think the paper is quite strong, but the paper would benefit from further clarification in the following sections:
- better description of the labelling process and if bias can be introduced there.
- better description of the protocol for qualitative validation and the order of assessment, and possible introduction of bias
- some analysis of the degree of inter-rater variability, how it has affected the results, and how it can be mitigated.

**Special Issue:**

yes

---

### Meta-Review · Area_Chair_BuaM · 2022-02-16

**Recommendation:** Accept (Oral)
**Confidence:** 4

**Metareview:**

The reviewers were quite enthusiastic about this paper, and I follow their suggestion and recommend acceptance. The main open question seems to have been whether ECG analysis belongs in the medical imaging community, but I agree with the authors and some of the reviewers that this paper is relevant and will find an interested audience in the MIDL community.

---

### Decision · Program_Chairs · 2022-02-28

Accept